# Evidence-Based Nutritional Recommendations for Maintaining or Restoring Nutritional Status in Patients with Amyotrophic Lateral Sclerosis: A Systematic Review

**DOI:** 10.3390/nu17050782

**Published:** 2025-02-24

**Authors:** Mariana Dantas de Carvalho Vilar, Karla Monica Dantas Coutinho, Sancha Helena de Lima Vale, Mario Emilio Teixeira Dourado Junior, Gidyenne Christine Bandeira Silva de Medeiros, Grasiela Piuvezam, Jose Brandao-Neto, Lucia Leite-Lais

**Affiliations:** 1Postgraduate Program in Health Sciences, Federal University of Rio Grande do Norte, Natal 59012-300, Brazil; marivilarnutri@gmail.com (M.D.d.C.V.); sancha.vale@ufrn.br (S.H.d.L.V.); gpiuvezam@yahoo.com.br (G.P.); brandao-neto@live.com (J.B.-N.); 2Laboratory of Technological Innovation in Health (LAIS), Federal University of Rio Grande do Norte, Natal 59012-300, Brazil; kmdcoutinho@gmail.com; 3Systematic Review and Meta-Analysis Laboratory-Lab-SYS/CNPq, Natal 59078-970, Brazil; gidyenne.silva@ufrn.br; 4Department of Nutrition, Federal University of Rio Grande do Norte, Natal 59078-970, Brazil; 5Department of Integrated Medicine, Federal University of Rio Grande do Norte, Natal 59012-300, Brazil; medourado03@gmail.com; 6Department of Public Health, Federal University of Rio Grande do Norte, Natal 59078-970, Brazil

**Keywords:** amyotrophic lateral sclerosis, nutritional assessment, nutrition therapy

## Abstract

**Background/Objectives**: This study is a systematic review of guidelines that aims to synthesize evidence-based recommendations to support appropriate nutritional management for patients with amyotrophic lateral sclerosis (ALS). **Methods**: PubMed/MEDLINE, Embase, Scopus, SciELO, Web of Science, LILACS, ScienceDirect, and Google Scholar were searched for records published up to July 2024. Clinical practice guidelines addressing any aspect of nutritional intervention in ALS were included. No language or country of publication restrictions were applied. Data extraction was performed by two independent reviewers. The methodological quality of the reports was assessed using the AGREE II instrument. Discrepancies were resolved by consensus. **Results**: The findings and main recommendations were summarized narratively. A total of 837 records were identified, and 11 were included in this review. The overall AGREE II scores for the included studies ranged from 3 to 7. The summary of nutritional recommendations was organized into topics: (1) dysphagia, (2) nutritional assessment, (3) energy, (4) protein, (5) supplementation, and (6) percutaneous endoscopic gastrostomy (PEG). This review summarizes relevant and updated nutritional recommendations to maintain or restore the nutritional status of patients with ALS, contributing to their quality of life and survival time. **Conclusions**: These nutritional recommendations will help health professionals and caregivers to implement and standardize nutritional care according to evidence-based practice in ALS. PROSPERO registration number CRD42021233088.

## 1. Introduction

Amyotrophic lateral sclerosis (ALS) is a progressive neurodegenerative disorder affecting both upper and lower motor neurons, leading to widespread motor impairment and cell death [1,2]. The global prevalence of ALS ranges from 1.57 to 9.62 cases per 100,000, while its incidence varies between 0.42 and 2.76 per 100,000 people per year. Higher prevalence and incidence rates are observed in developed countries [3]. Globally, ALS prevalence rises after the age of 50, peaks around the age of 85, and then declines [4]. However, rare early-onset cases can occur [4]. ALS is a severe condition with a median survival time of only 3 to 4 years following diagnosis [5,6,7,8].

Malnutrition is common in ALS patients, occurring in 16 to 53% of cases [9]. Body mass index (BMI) is a key indicator for assessing nutritional status, as a lower BMI is associated with accelerated disease progression and higher mortality risk in this population [10]. Marin et al. [11] reported that a 5% loss in body weight heightens the risk of death by 30% in ALS patients. Given these risks, maintaining proper nutrition is crucial for preserving functional capacity, enhancing quality of life, and potentially extending survival [12,13,14].

Multiple factors contribute to malnutrition in ALS, including impaired swallowing, loss of appetite, digestive disturbance, cognitive decline, lack of motivation, psychological conditions, and inadequate dietary intake. Additionally, an elevated metabolic rate may occur, potentially raising the risk of undernutrition or worsening this condition, particularly when adequate nutritional support is lacking [15,16]. Given these complexities, evidence-based nutrition interventions are essential and should be tailored to the stage of disease progression [17].

Evidence-based interventions and recommendations are typically summarized in clinical practice guidelines (CPGs), which serve as valuable tools for healthcare professionals to enhance and standardize care [18,19]. However, comprehensive guidelines addressing all aspects of nutritional management in ALS remain scarce. Existing recommendations primarily focus on dysphagia management and gastrostomy, while other critical nutritional aspects—including energy and nutrient requirement, texture-modified diets, nutritional supplementation—remain insufficiently explored.

To address this gap, it is essential to consolidate and analyze existing scientific evidence from protocols and clinical guidelines. A structured synthesis of nutritional recommendations could provide healthcare professionals with practical guidance, ultimately improving nutritional therapy and patient outcomes in ALS [20,21]. Therefore, this study aims to present an evidence-based summary of nutritional recommendations to support optimal dietary management and enhance the nutritional status of individuals living with ALS.

## 2. Materials and Methods

This systematic review was registered on the PROSPERO platform under number CRD42021233088, and the protocol was previously published [22]. This systematic review follows the Preferred Reporting Items for Systematic Reviews and Meta-Analyses [23], and the PRISMA checklist is provided (Appendix A). The participants, intervention, comparison, attributes of clinical practice guidelines and recommendations (PICAR) strategy for study selection was applied and is described in the study protocol [22].

### 2.1. Data Sources and Research Strategy

A systematic search was carried out up to July 2024 in the following databases: PubMed/MEDLINE, Embase, Scopus, SciELO, Web of Science, LILACS, ScienceDirect, and Google Scholar. For the search equations, we utilized Medical Subject Headings (MeSH) and Emtree terms, ensuring precise indexing. Boolean operators (AND and OR) were applied according to the equations established for each database. The equations were structured based on the PICAR acronym for recommendations and guidelines. Accordingly, we included the following descriptors: “Amyotrophic Lateral Sclerosis” [MeSH], “Motor Neuron Disease” [MeSH], “Nutrition Therapy” [MeSH], “Nutritional Assessment” [MeSH], “Diet” [MeSH], “Dietary Supplements” [MeSH], “Disorders of Swallowing” [MeSH], “Guidelines” [MeSH], “Practice Guideline” [MeSH], “Diet Therapy” [Emtree], “Dysphagia” [Emtree] and “Practice Guideline” [Emtree]. The complete search strategies are provided (Appendix A).

### 2.2. Selection Criteria 

For this study, the inclusion criteria were clinical practice guidelines that addressed any aspect of nutritional therapy to maintain or restore nutritional status in patients diagnosed with definite, probable, or possible ALS. Guidelines focusing on adults (aged 18 years or older) and elderly individuals of both sexes with any of the aforementioned clinical diagnoses were included. Exclusion criteria were guidelines with other neurodegenerative diseases that did not include ALS or that did not address nutritional therapy in ALS. No restrictions on language or country of publication were applied.

### 2.3. Outcome Measures

A summary of nutritional recommendations to maintain or restore the nutritional status of ALS patients was performed.

### 2.4. Study Selection

For all identified studies, at least two researchers (MDCV and LLL) independently screened and reviewed the titles and abstracts using the Rayyan QCRI^®^ tool. Articles that met the inclusion criteria were reviewed in full. Any disagreements were resolved in discussion with a third reviewer (SHLV). A manual search was performed to search for guidelines that were not found through the search equations performed in the electronic databases and libraries. Thus, the manual search found and included the website of the Brazilian Ministry of Health, mentioned as an organization in the flow diagram. Information about the phases of this process is described through the PRISMA flow diagram (Figure 1).

### 2.5. Data Extraction

Data extraction was performed in a standardized way, using Microsoft Excel^®^ by three independent researchers (MDCV, KMDC, and LLL). Unclear information or discrepancies between data extraction were resolved by consensus. The characteristics of the studies were related according to the research question. The following data were extracted: general information about the guideline (title, authors, or organization, year of publication, and funding), nutritional recommendations provided, and the stratification of the level of evidence used. No study reports contained incomplete or missing data; therefore, corresponding authors were not contacted.

### 2.6. Assessment of Guideline Quality

Two independent authors (KMDC and MDCV) assessed the methodological quality of the reports using the Appraisal of Guidelines for Research & Evaluation (AGREE) II instrument [24] and then discussed it with the other authors to reach a final score (Appendix A). This tool comprises 23-items organized into six domains: 1. Scope and purpose, 2. Stakeholder engagement, 3. Rigor of development, 4. Clarity of presentation, 5. Applicability, and 6. Editorial independence. The final component of the AGREE II evaluation involves rating for the overall quality of the report from 1 (lowest quality) to 7 (highest quality) [24]. A score of 1 (Strongly Disagree) was assigned when the AGREE II item was very poorly reported or did not meet the criteria. Scores ranging from 2 to 6 were given when the reporting of the AGREE II item did not fully meet the criteria or considerations. Higher scores indicated better alignment with the criteria and considerations. A score of 7 (Strongly Agree) was given when the quality of reporting was exceptional and met all the criteria and considerations in full [24].

### 2.7. Data Synthesis

For this systematic review, the findings and main recommendations were narratively summarized.

## 3. Results

### 3.1. Research Resources and Guidelines

The selection process is shown in Figure 1. The search strategy identified 837 records across multiple databases: PubMed (n = 35), EMBASE (n = 60), Scopus (n = 30), SciELO (n = 261), LILACS (n = 59), Science Direct (n = 198), Web of Science (n = 20), Google Scholar (n = 174). The Rayyan QCRI^®^ tool automatically removed 65 duplicates, and 364 studies were removed after reviewing titles and abstracts. A guideline from the Brazilian Ministry of Health was included through a manual search (n = 1). Following the screening process, 11 records were included in this review, all of which were clinical practice guidelines from various countries, as outlined in Table 1. The nutritional recommendations from these guidelines are provided in Table 2.

### 3.2. Quality of Guidelines

The characteristics of each study are detailed in Table 1, including the AGREE II score. All studies conducted a systematic literature search to support their recommendations. The data extracted and the synthesis of information varied across studies. They are further described narratively. Most studies classified the strength of recommendations as either strong or weak, while some rated the level of evidence as high, moderate, low, or very low.

### 3.3. Summary of Recommendations

The synthesis of information and the main nutritional recommendations were organized into six topics: (1) dysphagia, (2) nutritional assessment, (3) energy, (4) protein, (5) supplementation, and (6) percutaneous endoscopic gastrostomy (PEG). The recommendations are described in Table 2 and summarized in Figure 2.

### 3.4. Nutritional Evaluation

Weight and muscle mass loss affect a large proportion of patients (15–55%) as the disease progresses, resulting in a reduction in the body mass index (BMI) and contributing to malnutrition. Malnutrition negatively impacts the prognosis of ALS and often results in loss of appetite, low food intake, dysphagia, apathy, and hypermetabolism [36]. Studies show that maintaining an appropriate weight for height can positively influence on ALS outcomes, increasing life expectancy compared to patients who experience weight loss and malnutrition [37]. 

Thus, adequate nutritional assessment and appropriate care are crucial in the management of patients with ALS [37]. Key measures in nutritional assessment for ALS patients include weight, height, dual-energy X-ray absorptiometry (DEXA), and/or bioelectrical impedance analysis (BIA) with a validated formula (if available). Clinical guidelines recommend that nutrition professionals calculate BMI, monitor weight loss over time, and assess body composition to develop a more tailored dietary plan based on the patient’s current nutritional status [29,38].

Weight, nutritional status, and swallowing should be regularly assessed at each visit or every 3 months by the nutritionist and/or multidisciplinary team. This helps identify possible causes of nutritional deficits, such as weight loss, loss of appetite, reduced food and liquid intake, and disease-related factors, including dysphagia, insufficient respiratory capacity, psychological disorders, muscle atrophy and weakness [27,29,30,32,34,35].

### 3.5. Dysphagia

Dysphagia significantly impacts nutritional intake and hydration, contributing to the deterioration of nutritional status in ALS patients as the disease progresses [29]. Patients with dysphagia are at greater risk of aspiration of liquids and/or foods of varying consistencies, which can lead to pulmonary infections [39]. Guidelines suggest that for mild to moderate dysphagia, transitioning from a normal to a soft diet is the first multidisciplinary recommendation, as foods with a pureed consistency and lower residue are safer for these patients [29,30,32].

In addition to a multidisciplinary approach, nutritionists and speech–language pathologists should assess oral intake and dysphagia symptoms at each visit to make necessary adjustments. This may include modifying food consistency and preparation, encouraging smaller, more frequent meals, using oral nutritional supplements (ONS), adjusting posture during meals, eating slowly, employing swallowing techniques such as the chin tuck maneuver, and ensuring proper oral hygiene [25,26,29,30,32,33,35].

For patients with severe dysphagia and progressive weight loss, alternative feeding methods, such PEG, may be required to reduce the risk of choking while ensuring adequate nutrition and hydration. When feasible, enteral tube feeding should complement, rather than replace, oral feeding. However, it provides a safer and more convenient method of delivering food, medications, and water, helping to maintain or restore their nutritional and hydration status in ALS patients [26,29,30,34,40].

### 3.6. Energy

Indirect calorimetry (IC) is considered the gold standard for measuring resting energy expenditure (REE) in ALS patients. However, despite supporting evidence, IC is seldom used in clinical practice due to its high cost and limited accessibility for healthcare professionals. In non-ventilated ALS patients, energy requirements should be estimated at 30–35 kcal/kg of body weight, depending on physical activity and the progression of nutrition status. Alternatively, predicted equations such as Harris–Benedict or Mifflin–St Jeor formulas can be used. For patients with non-invasive ventilation, whose REE is typically lower, energy requirements should be estimated at 25–30 kcal/kg of body weight [29,33].

In contrast, the guideline for the Japanese population with ALS recommends a high-calorie, high-fat diet and proposes two specific formulas for calculating energy intake at an early stage, which are calculated as follows: (1) TEE = 1.68 × BEE + 11.8 × ALSFRS-R – 690; (2) REE = 1.000251 × BEE + 313.3507 × TV − 112.036. These formulas account for total energy expenditure (TEE), the functional scale (ALSFRS-R), basal energy expenditure (BEE) calculated using the Harris–Benedict formula, tidal volume (TV), and the physical activity factor [34].

Daily energy intake should be adjusted to meet the changing needs of the patient as the disease progresses [25,29]. A high-calorie diet helps maintain or restore nutritional status and may also increase survival. Additionally, a high-calorie, high-carbohydrate diet may be more beneficial than a high calorie, high-fat diet [30,33].

### 3.7. Protein

There are no specific recommendations for the ideal amount of protein for ALS patients. Key factors such as age, renal function, level of stress, nutritional status, hypercatabolic conditions, and disease progression should be considered when determining and adjusting daily protein intake. According to the guidelines, protein requirements can range from normal to high, typically between 0.8 and 1.5 g/kg/day [25,29].

### 3.8. Supplementation

Nutritional supplementation is recommended for ALS patients only when their dietary intake does not meet their nutritional needs. The prescription of high-calorie, high-protein supplements should be evaluated based on the patient’s complaints of fatigue, effortful eating, weight loss, and overall nutritional status during follow-up. Indicators such as weight, BMI, and other anthropometric and biochemical parameters can guide the need for supplementation [29,32,35].

A similar approach applies to the supplementation of micronutrients and bioactive compounds. There is insufficient evidence to support the supplementation of these compounds in ALS patients without specific deficiencies, as it has not been shown to positively impact survival [26,32]. Therefore, it is recommended that ALS patients achieve at least the Recommended Daily Intake (RDI) of micronutrients, as established for the general population, either through diet or supplementation [38].

### 3.9. Percutaneous Endoscopic Gastrostomy (PEG)

PEG is a clinically safe and effective method that provides valuable access for delivering nutrition and hydration, helping to maintain body weight in patients with ALS [28]. After PEG placement, enteral tube feeding can be exclusive or, when feasible, combined with an oral diet. PEG has been shown to be superior to percutaneous radiological gastrostomy (PRG) in terms of complications related to gastrostomy tube insertion. As a result, PEG is the preferred modality for ALS patients over other alternative feeding and nutrition routes [13].

Determining the optimal time for PEG tube insertion presents a challenge for the multidisciplinary team and must be tailored to each patient. Transitioning from oral feeding to a tube-feeding route is a significant decision, especially for patients who still enjoy the taste and social aspects of mealtimes. Once the decision is made, it is recommended that PEG insertion occur within 4 weeks [25,27,30,34]. However, discussions about PEG should begin early in the disease process and continue regularly as ALS progresses, considering the increased risk of aspiration during swallowing. Discussions should focus on the benefits of early PEG insertion versus the risks of delaying the procedure. Signs such as dysphagia, delayed meal duration, weight loss, worsening respiratory function, and the risk of choking should guide the team’s decision to recommend and perform PEG [30,34,35].

Additionally, guidelines suggest that a decrease in forced vital capacity (FVC) to 50% should be considered a key indicator for the placement of an enteral tube placement, even in the absence of dysphagia. An FVC <50% should not rule out the recommendation for PEG, as long as respiratory status is carefully monitored during and after the procedure [30,32,34]. Early PEG insertion is often beneficial for stabilizing weight over the long term, preventing malnutrition, and supporting ongoing nutritional maintenance. These factors directly contribute to improved quality of life and survival [25,26,29,31,35].

## 4. Discussion

This systematic review summarizes the recommendations from clinical practice guidelines on nutritional care for ALS patients. The recommendations cover various aspects, including nutritional assessment, daily energy and nutrient requirements, dysphagia management, the need for supplementation, and alternative feeding methods. The included guidelines provided general nutritional recommendations with varying levels of evidence, sometimes relying on expert consensus when evidence was insufficient. However, the majority of recommendations were based on primary studies, clinical trials, or systematic reviews, offering a higher level of reliability for guiding clinical practice. 

Furthermore, there is limited evidence on the role of other healthcare professionals, such as nurses, psychologists, and dentists, in supporting nutritional care, despite evidence showing that multidisciplinary care positively impacts survival in ALS [41]. Additionally, there is a lack of data regarding the optimal number of professionals involved in patient care, which specific professionals should be included on the team, the necessary equipment for care environment, and how nutritional monitoring should be conducted in the home setting [42].

Studies indicate that factors as loss of appetite, sialorrhea, bulbar symptoms, and weight loss contribute to reduced BMI in ALS patients, which is associated with reduced survival and a higher risk of mortality [10,29,43,44]. Therefore, providing specific nutritional support and management tailored to each stage of the disease, based on scientific evidence, is crucial to ensuring consistent quality of care. This approach positively impacts the patient’s quality of life and survival [45].

Among all included studies, the AGREE II domains with the highest scores were “scope and purpose”, “stakeholder involvement”, and “clarity of presentation”. The lowest score was for the item “a procedure for updating the guideline is provided”, indicating that few guidelines offer comprehensive information on how their recommendations were formulated and updated. This highlights a gap in ensuring that some guidelines remain current for guiding healthcare professionals in clinical practice over time.

Among the eleven studies included, six (55%) did not specify or provide recommendations on macronutrient distribution ranges or micronutrient and bioactive compounds supplementation. This absence may be due to the limited focus of nutritional management in ALS, which primarily emphasizes energy intake requirements and alternative feeding routes. Additionally, the lack of clinical trials and systematic reviews on the effects of nutritional supplementation as a complementary therapy further contributes to this gap.

Despite certain weaknesses in specific items of the included guidelines (AGREE II score, Table 1), the strength of the available evidence remains significant. This evidence supports the most relevant and up-to-date nutritional recommendations for ALS patients. The summarized recommendations (Figure 2) provide clear, specific, and evidence-based guidance for healthcare professionals managing the nutritional needs of these patients. Additionally, they enhance the development of more effective nutritional care approaches for this population.

Broad and updated nutritional recommendations for ALS patients is crucial, as it can help reduce nutritional risk and positively impact both quality of life and survival. These findings have positive implications not only for clinical practice but also for healthcare policies and future research. Further studies and collaborative efforts are essential to refine these recommendations, ensure their implementation in clinical settings, and address existing gaps in nutritional management for ALS patients. A limitation of this review was the inability to perform a meta-analysis due to the nature of the included studies, restricting the results to a narrative synthesis.

## 5. Conclusions

This systematic review highlights current nutritional recommendations for ALS patients, covering six key aspects of care: nutritional status, dysphagia, energy, protein, supplementation, and PEG. The summarized recommendations reflect diverse perspectives from different countries, including approaches to nutritional assessment, body composition analysis, energy requirement estimation, protein intake guidelines, dysphagia management, and the timing of PEG as an alternative feeding route. Despite variations in the strength of evidence, most recommendations are supported by clinical trials and systematic reviews, reinforcing their relevance in clinical practice. However, gaps remain, particularly regarding the role of multidisciplinary care, optimal team composition, and the integration of nutritional monitoring in home settings. Additionally, there is a need for more high-quality evidence to determine precise energy and protein requirements for different disease stages, as well as the impact of specific dietary interventions on disease progression and survival. The lack of standardized guidelines for macronutrient distribution and supplementation further highlights the need for further research to identify optimal nutrient ratios, evaluate the effectiveness of different supplementation strategies, and establish clear nutrition requirements for ALS population.

This review underscores the need for continuous updates to nutritional guidelines to improve the quality of care and survival of ALS patients. Integrating broad, evidence-based recommendations into clinical practice and healthcare policies can enhance patient outcomes. Future research should focus on refining these guidelines, conducting large-scale longitudinal studies to assess long-term nutritional interventions, and exploring personalized nutrition approaches tailored to disease progression and individual metabolic needs. Expanding multidisciplinary collaboration and conducting high-quality studies are crucial steps toward optimizing ALS care worldwide.

## Figures and Tables

**Figure 1 nutrients-17-00782-f001:**
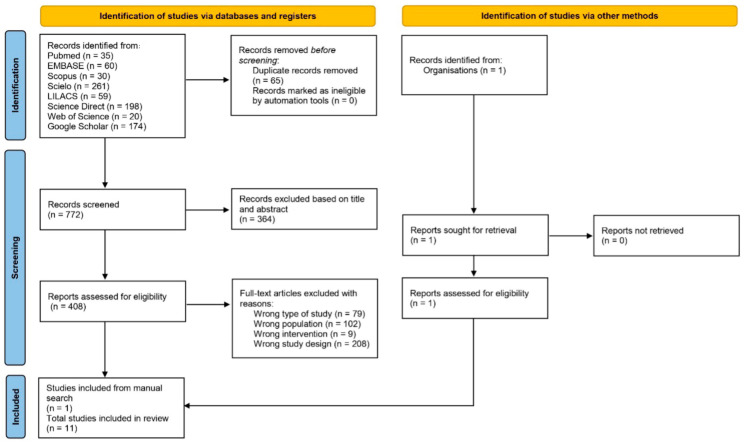
PRISMA flow diagram.

**Figure 2 nutrients-17-00782-f002:**
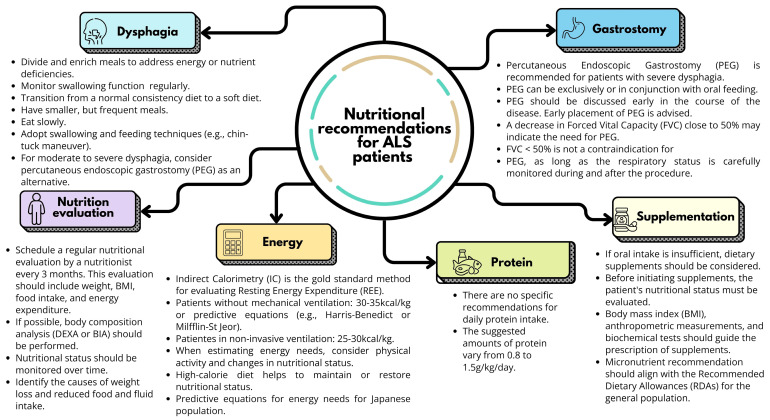
Summary of nutritional recommendations for patients with amyotrophic lateral sclerosis. Figure of own authorship.

**Table 1 nutrients-17-00782-t001:** Characteristics of the studies included.

Author (Year) [Ref]	Country, Region or Society	Language	Nutrition Topics Addressed	AGREE II
Heffernan et al. (2004) [25]	United Kingdom	English	Nutritional status, Dysphagia, Energy, Supplementation, PEG	5
Miller et al. (2009) [26]	American Academy of Neurology	English	Supplementation, PEG	6
Andersen et al. (2012) [27]	European Federation of Neurological Society	English	Dysphagia, PEG	5
Oliver et al. (2017) [28]	National Institute for Health and Care Excellence	English	Nutritional status, Dysphagia, PEG	5
Burgos et al. (2018) [29]	European Society for Clinical Nutrition and Metabolism	English	Nutritional status, Dysphagia, Energy, Protein, Supplementation, PEG	6
Shoesmith et al. (2020) [30]	Canada	English	Nutritional status, Dysphagia, Energy, PEG	7
BRASIL (2021) [31]	Brazil	Portuguese	PEG	6
Boostani et al. (2023) [32]	Iran	English	Nutritional status, Dysphagia, Supplementation, PEG	5
Petri et al. (2023) [33]	German Society of Neurology	German	Dysphagia, Energy, PEG	4
Urushitani et al. (2023) [34]	Japan	Japanese	Nutritional status, Dysphagia, Energy, Supplementation, PEG	6
Van Damme et al. (2024) [35]	European Academy of Neurology	English	Nutritional status, Dysphagia, PEG	5

Abbreviation: AGREE II, Appraisal of Guidelines for Research & Evaluation II. Ref, Reference. PEG, percutaneous endoscopic gastrostomy.

**Table 2 nutrients-17-00782-t002:** Guideline content related to nutrition in amyotrophic lateral sclerosis.

Authors (Year) [Ref]	Dysphagia	Nutrition Evaluation	Supplementation	Energy	Protein	PEG
Heffernan et al. (2004) [25]	Multidisciplinary therapy and continuous assessment. Modification of food consistency and monitoring of swallowing. Safe feeding techniques (e.g., chin tuck). Education for patients and caregivers on swallowing and feeding techniques. Involvement of occupational or physical therapy	NR	Vitamins and minerals should be obtained through the diet rather than supplements	Energy needs should be regularly monitored, and intake should be adjusted to meet requirements as the disease progresses	NR	PEG helps stabilize weight, prevent malnutrition, and maintain long-term nutritional status. Individualized approach is required for the timing of PEG placement. PEG is not recommended in the advanced stages of the disease. PRG is an alternative when PEG is not indicated. NGT feeding is considered for short-term use
Miller et al. (2009) [26]	NR	NR	Creatine (5–10 g/day): Does not slow disease progression. Vitamin E: High doses should not be considered as a treatment, and there is no recommendation for low doses	NR	NR	PEG stabilizes body weight when oral feeding is no longer sufficient. There is insufficient evidence to support or refute the optimal timing for PEG insertion. PEG may help prolong survival
Andersen et al. (2012) [27]	Referral to a nutritionist and a speech therapist can provide guidance on appropriate swallowing techniques	The evaluation of nutritional status, including body weight measurement, should be performed at each visit	NR	NR	NR	When considering PEG insertion, consider bulbar symptoms, malnutrition (weight loss > 10%), respiratory function, and the patient’s overall health. Early insertion of PEG is recommended, and patients and caregivers should be made aware of the possible risks and benefits. PRG is a suitable alternative to PEG. NGT is considered for short-term use. Home parenteral nutrition is indicated in advanced stages
Oliver et al. (2017) [28]	The patient’s ability to eat and drink should be assessed. Assistance should be provided during feeding. The consistency of food and liquids should be modified as needed. Guidance on proper positioning during meals should also be given	The patient’s weight, nutritional status, and swallowing ability should be assessed	NR	NR	NR	PEG should be discussed early and regularly. When indicated, it should be performed promptly without delay
Burgos et al. (2018) [29]	Break up and enrich meals to provide additional energy or address nutrient deficiencies. Assess the need for oral nutritional supplementation in cases of progressive weight loss	Complete nutritional assessment should include BMI, weight loss, and lipid status. If available, body composition analysis using DEXA or BIA should be performed. Ongoing monitoring of nutritional status should focus on BMI and weight changes over time. Weight gain is recommended for patients with BMI < 25.0 kg/m^2^. For patients with BMI between 25 and 35 kg/m^2^, weight stabilization should be targeted. For patients with BMI > 35 kg/m^2^, weight loss is recommended to enhance both passive and active mobilization	Nutritional supplementation is recommended when a patient’s nutritional needs are not fully met through diet alone	Energy requirements for non-ventilated patients should be estimated at 30 kcal/kg of body weight, considering physical activity and weight changes. For patients with non-invasive ventilation, energy needs range from 25 to 30 kcal/kg of body weight, or can be calculated using the Harris–Benedict equation	There are insufficient data to make specific recommendations. However, factors such as age, kidney function, and level of stress should be considered when determining nutritional needs	Discussions about PEG should occur early and be revisited regularly. PEG should be considered in the presence of dysphagia, prolonged meal duration, weight loss, impaired respiratory function, choking risk, and indivudual’s preferences. It is recommended to consider PEG before substantial weight loss and worsening respiratory function. Patients and caregivers should be fully informed about possible risks and benefits. Enteral nutrition should be prioritized, with parenteral nutrition considered only when enteral feeding is not viable. Home parenteral nutrition is generally not indicated
Shoesmith et al. (2020) [30]	Swallowing function should be regularly assessed and monitored	Weight and BMI should be monitored every three months or as clinically indicated	NR	High-calorie diets may help improve nutritional indicators and enhance survival. High-calorie, high-carbohydrate diets may be more beneficial than high-calorie, high-fat diets	NR	Factors to indicate PEG are heightened risk of aspiration despite changes in diet texture or compensatory strategies, weight loss of ≥5–10%, a reduction of ≥1 point in usual/baseline BMI, BMI < 18.5, or TDEE exceeding daily energy intake. Patients should be informed of the risks and benefits PEG should be performed within four weeks after a shared decision between the healthcare team and the patient. The nutritionist should regularly monitor the prescribed enteral feeding. NGT is not recommended for long-term use, and parenteral nutrition should be considered only in exceptional cases
BRASIL (2021) [31]	NR	NR	NR	NR	NR	PEG helps stabilize or increase body weight, prolong survival, and improve quality of life
Boostani et al. (2023) [32]	An initial speech and language pathology consultation should be conducted. Strategies for modifying food consistency, implementing postural adjustments, and using swallowing maneuvers should be applied. PEG should be considered in more severe stages of dysphagia, and high-viscosity liquids should be handled with caution. Oral hygiene should be maintained throughout the day	A regular nutritional assessment should be conducted by a nutritionist every 3 months. It is recommended to assess nutritional history, perform a clinical examination evaluating swallowing, weight, and BMI. The assessment of body composition and energy expenditure may be considered on an individual basis	Nutritional counseling should include food fortification, oral nutritional supplementation, and the potential need for early enteral nutrition, such as PEG. When weight loss, fatigue, or effortful eating is present, food fortification is recommended. Oral nutritional supplementation is advised for patients with unmet energy needs	NR	NR	PEG should be considered in cases of worsening speech or changes in food consistency, weight loss of 10%, BMI < 18.5 kg/m^2^, risk of bronchoaspiration, or if FVC > 50%. PEG and NGT have similar efficacy in maintaining food intake, but PEG is superior in quality of life measures
Petri et al. (2023) [33]	The speech and language pathologist plays a key role in detecting clinically inapparent dysphagia and adjusting treatment accordingly. Regular monitoring for dysphagia is recommended	Patients should be regularly monitored for weight loss	In situations involving patient discomfort, weight loss, dehydration, and risk of aspiration, the nutritional counseling may include the prescription of high-calorie liquid nutrition	A high-calorie diet can be beneficial, especially in patients experiencing rapid disease progression	NR	PEG is recommended for patients with advanced dysphagia and significant weight loss
Urushitani et al. (2023) [34]	Swallowing function should be monitored regularly. Screening tests for dysphagia include repetitive saliva swallowing, modified water swallowing, and cervical auscultation. Swallowing rehabilitation (direct and indirect) by a multidisciplinary team is essential. Expiratory strength training is recommended to maintain swallowing function. In cases of risk of choking and/or aspiration, PEG should be considered. Surgical intervention for aspiration prevention may also serve as an alternative approach	Weigh monitoring is important, as weight loss and reduction rate of weight are independent prognostic factors for short survival	NR	The formulas for calculating the ideal energy intake for early-stage Japanese patients are: (1) TEE = 1.68 × BEE + 11.8 × ALSFRS-R − 690, and (2) REE = 1.000251 × BEE + 313.3507 × TV − 112.036. A high-calorie, high-fat diet is recommended to maintain body weight and prolong survival. In patients using NIV, energy intake should be restricted. For patients in a fully blocked state, the daily energy requirement is less than 800 kcal/day	NR	PEG should be considered when there is a risk of aspiration and progressive weight loss. Parenteral nutrition may be considered when other feeding routes are not viable. Indications for PEG include a >10% reduction in premorbid weight, early dysphagia, reduced and delayed food intake, and preserved respiratory function (FVC ≥ 50%). PEG should ideally be performed before arterial carbon dioxide pressure increases. Patients with low FVC or those on NIV can undergo PEG with respiratory support. PRG is not widely used in Japan
Van Damme et al. (2024) [35]	If there is weight loss or swallowing difficulties, consult a nutritionist, speech therapist, and/or occupational therapist. Consider the composition of food, food consistency, frequency of meals, intake and consistency of liquids, risk of choking, use of utensils, and optimal positioning and seating	Identify the causes of weight loss and reduced food and fluid intake, such as swallowing difficulties, respiratory failure, depression, loss of appetite, muscle atrophy, and upper limb weakness	Consider the use of dietary supplements in cases of weight loss or swallowing difficulties	NR	NR	Discuss PEG at an early stage and at regular intervals. Explain the benefits of early insertion and the risks associated with late insertion. In patients with respiratory failure, the use of NIV during PEG insertion is recommended. When PEG is indicated, it should be performed without delay. Consider NGT while awaiting PEG insertion. Engage family members and/or caregivers in discussions about PEG. If PEG is not feasible, consider TPN feeding as an option

Abbreviations: ALSFRS-R: ALS functional rating scale-revised; BEE: basal energy expenditure calculated by the Harris–Benedict formula (kcal/day); BIA: bioelectrical impedance analysis; BMI: body mass index; NGT: nasogastric; NR: No recommendation; PEG: percutaneous gastrostomy; PRG: percutaneous radiologic gastrostomy; DEXA: dual-energy X-ray absorptiometry; FVC: forced vital capacity; Ref, Reference; TDEE: total daily energy expenditure; REE: resting energy expenditure (kcal/day); TEE: total energy expenditure (kcal/day); TV: tidal volume (L).

## Data Availability

No new data were created or analyzed in this study. Data sharing is not applicable to this article.

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
