# Peer review of "Evidence-Based Nutritional Recommendations for Maintaining or Restoring Nutritional Status in Patients with Amyotrophic Lateral Sclerosis: A Systematic Review"

_nutrients, 2025, doi:10.3390/nu17050782_

Round 1
Reviewer 1 Report
Comments and Suggestions for Authors
thanks for the opportunity to score this ms. It deals with an important topic with clinical hints. it looks that systematic reviews on the topic are missing. This is due to the low number of reliable works on several aspects. Also in this case the reader has the idea that the number of work available are not sufficient to draw conclusions and large part of the results section appears largely speculative appearing, more as narrative than systematic. How is possible to draw reccomendation based on very few published works? I would avoid discussing the domains supplementation, energy and protein and try to strictly stay on evidences for the other domains. Indeed, as proposed, discussion is not a discussion of the evidences but rather a short description of the importance to have raccomendations. Instead a proper discussion of the single domains with open issues should be implemented.
Author Response
Please see the answers to Reviewer 1 attached.

Reviewer 2 Report
Comments and Suggestions for Authors
Review of the Manuscript Entitled: Evidence-Based Nutritional Recommendations for Maintaining or Restoring Nutritional Status in Patients with ALS: A Systematic Review
The impact of diet on the development of various neurodegenerative diseases is a crucial area of research. Therefore, the topic addressed by the authors is highly relevant. However, I have identified several issues that need to be addressed, and numerous improvements should be made.
- The abbreviation "ALS" should be defined in the title.
- While the introduction is well-structured, it appears to be entirely plagiarized. Specifically, lines 44 to 80 (the entire introduction) are identical to the content available at https://bmjopen.bmj.com/content/12/8/e064086.full. Copying text, even from one's own publications, is considered unethical and constitutes academic misconduct. If the authors wish to proceed with publication, they must thoroughly revise and rewrite the introduction.
- The description of the methodology is appropriate.
- If possible, the readability of Figure 1 should be improved, as the current resolution appears to be low.
- The structure of Tables 1 and 2 is understandable; however, an additional column indicating reference numbers should be included to align with the journal's requirements.
- The results section appears to be well-documented, with appropriate citations from the literature.
- The discussion is relatively brief, with much of the information integrated into the results section. In the conclusions, the authors should provide a broader perspective on future research directions—what aspects require further study, what gaps remain in the current knowledge, etc.
Overall, while the manuscript covers an important topic, significant revisions are necessary to meet ethical and academic standards.
Author Response

(The authors gave the same response as above.)

Round 2
Reviewer 2 Report
Comments and Suggestions for Authors
I am satisfied with the changes made. The manuscript can be accepted.